# Proteome dynamics during homeostatic scaling in cultured neurons

**Aline Ricarda Dörrbaum[1,2], Beatriz Alvarez-Castelao[1], Belquis Nassim-Assir[1], Julian D Langer[1,3]\*, Erin M Schuman[1]\***

[1]Max Planck Institute for Brain Research, Frankfurt, Germany; [2]Goethe University Frankfurt, Faculty of Biological Sciences, Frankfurt, Germany; [3]Max Planck Institute of Biophysics, Frankfurt, Germany

**Abstract** Protein turnover, the net result of protein synthesis and degradation, enables cells to remodel their proteomes in response to internal and external cues. Previously, we analyzed protein turnover rates in cultured brain cells under basal neuronal activity and found that protein turnover is influenced by subcellular localization, protein function, complex association, cell type of origin, and by the cellular environment (Dörrbaum et al., 2018). Here, we advanced our experimental approach to quantify changes in protein synthesis and degradation, as well as the resulting changes in protein turnover or abundance in rat primary hippocampal cultures during homeostatic scaling. Our data demonstrate that a large fraction of the neuronal proteome shows changes in protein synthesis and/or degradation during homeostatic up- and down-scaling. More than half of the quantified synaptic proteins were regulated, including pre- as well as postsynaptic proteins with diverse molecular functions.

## Introduction

Long-lasting changes in synaptic strength, which are a basis for learning and memory formation, require de-novo protein synthesis as well as the degradation of existing proteins (*Cajigas et al., 2010*; *Hegde, 2017*; *Jarome and Helmstetter, 2013*; *Tai and Schuman, 2008*). For example, injections of the protein synthesis inhibitor puromycin (*Flexner et al., 1963*) or of the proteasome inhibitor lactacystin (*Lopez-Salon et al., 2001*) into rodent brains during specific time windows after training blocked long-term memory formation. Also, several forms of synaptic plasticity studied in vitro, including homeostatic scaling, require protein synthesis and degradation (*Ehlers, 2003*; *Kang and Schuman, 1996*; *Rosenberg et al., 2014*; *Schanzenbächer et al., 2016*). Homeostatic scaling is a compensatory mechanism that enables neurons to adjust the synaptic strength in response to persistent changes in the network activity. While homeostatic down-scaling reduces synaptic strength, at least in part by AMPA (α-amino-3-hydroxy-5-methyl-4-isoxazolepropionic acid) receptor (AMPAR) internalization, the opposite is true for homeostatic up-scaling, which increases synaptic strength by increased AMPAR surface expression (*Turrigiano, 2012*; *Turrigiano, 2008*; *Turrigiano et al., 1998*). In cultured neurons, distinct sets of proteins are differentially regulated during homeostatic scaling that is induced by pharmacological manipulation of neuronal activity. However, homeostatic scaling cannot be evoked in the presence of protein synthesis inhibitors or proteasome inhibitors (*Ehlers, 2003*; *Schanzenbächer et al., 2016*).

These findings suggest that information is stored in the specific combination of proteins that are present in neurons, especially at synapses. However, although memories can last for a lifetime, most neuronal proteins are continuously turned over and only a small fraction of neuronal proteins have been reported to be extremely long-lived (*Toyama et al., 2013*). In-vivo studies reported average protein half-lives of 9.0–10.7 days in the mouse brain (*Fornasiero et al., 2018*; *Price et al., 2010*), whereas protein half-lives determined in culture are comparably shorter (on average 2.3–5.9 days)

**\*For correspondence:**
julian.langer@brain.mpg.de (JDL);
erin.schuman@brain.mpg.de
(EMS)

**Competing interests:** The authors declare that no competing interests exist.

(*Cohen et al., 2013*; *Dörrbaum et al., 2018*; *Mathieson et al., 2018*) but correlate well with the in vivo data (*Fornasiero et al., 2018*). Continuous protein turnover is essential to maintain a functional pool of proteins and impaired protein degradation is implicated in various neurodegenerative diseases (*Boland et al., 2018*).

What molecular changes underlie homeostatic scaling in neurons and how are these changes accomplished? Previous studies used candidate-based approaches to show the differential expression during synaptic scaling of selected proteins, such as Bdnf (*Rutherford et al., 1998*) or PSD-95 (*Sun and Turrigiano, 2011*). Mass spectrometry (MS)–based proteomics allows for the untargeted analysis of all (detectable) proteins without prior candidate selection. Schanzenbächer et al. used BONCAT-MS to analyze the nascent proteome during homeostatic scaling and reported that hundreds of proteins showed regulated synthesis rates (*Schanzenbächer et al., 2018*; *Schanzenbächer et al., 2016*). In addition to the differential expression of new proteins, proteome remodeling can also be accomplished by the selective stabilization or removal of existing proteins. In addition, regulated protein synthesis rates might not change protein abundance levels, but protein turnover rates instead, when accompanied by changes in protein degradation. The contributions of protein synthesis and degradation to changes in protein abundance levels and protein turnover rates during synaptic plasticity have not yet been systematically studied. Here, we used dynamic SILAC (<u>S</u>table <u>I</u>sotope <u>L</u>abeling with <u>A</u>mino acids in <u>C</u>ell culture) in combination with MS, which enables proteome-wide analysis of protein synthesis and degradation from the same sample and allows one to quantify changes in synthesis and degradation across different experimental conditions. Our data represent the most comprehensive analysis of proteome dynamics associated with homeostatic scaling to date, revealing that large fractions of the neuronal proteome, especially synaptic proteins, are affected by this scaling. Most proteins exhibited a decrease in synthesis or degradation, whereas only a few proteins showed increased synthesis rates.

## Results

### Experimental design used to quantify changes in protein synthesis, degradation, turnover, and abundance

Dynamic SILAC labeling (*Figure 1A*) in combination with MS was used to quantify the protein dynamics during homeostatic scaling. Primary hippocampal neurons were grown in a 'light' medium that contained all natural amino acids. After 18–19 days in vitro (DIV), the medium was exchanged to a 'heavy' medium that contained heavy isotopically labeled arginine and lysine instead of the 'light' isotopologues. Upon the medium change, nascent proteins mainly incorporated 'heavy' arginine and lysine so that newly synthesized proteins became 'heavy' labeled, whereas pre-existing proteins were 'light' labeled. Together with the heavy SILAC pulse, drugs were added to elicit homeostatic scaling. Neurons were either treated with the GABA$_A$-receptor antagonist bicuculline (BIC), to increase global network activity and induce homeostatic down-scaling, or with the sodium channel blocker tetrodotoxin (TTX), to decrease neuronal network activity and induce homeostatic up-scaling (*Turrigiano et al., 1998*) or not treated (Ctrl). The cells were treated and heavy labeled for 1, 3 or 7 days. The heavy and light peptide counterparts were subsequently distinguished and quantified by MS. As demonstrated previously (*Dörrbaum et al., 2018*), heavy peptides could be robustly detected and quantified even after 1 day of heavy SILAC pulse (*Figure 1—figure supplement 1*).

To make sure that the treatments were effective and induced homeostatic scaling over the entire duration of the experiment, we measured miniature excitatory postsynaptic currents (mEPSCs) after 7 days of each treatment. Previous studies demonstrated that a 24-hr BIC treatment (down-scaling) elicited decreased mEPSC amplitudes and that a 24-hr TTX treatment (up-scaling) elicited increased mEPSC amplitudes (*Schanzenbächer et al., 2016*; *Turrigiano et al., 1998*). Consistently, decreased or increased mEPSC amplitudes were also observed after 7 days of BIC treatment (*Figure 1—figure supplement 2A–B*) or TTX treatment (*Figure 1—figure supplement 2C–D*), respectively, demonstrating that the treatments induced homeostatic scaling over the entire duration of the experiment.

As described previously (*Dörrbaum et al., 2018*), protein turnover rates were determined ousing the fraction of remaining pre-existing protein (*Figure 1B*) over time. The turnover rates of all

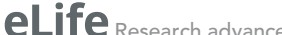

**Figure 1.** Experimental design to quantify protein synthesis and degradation during homeostatic scaling. (**A**) Dynamic SILAC experiment. Primary hippocampal cells were grown for 18–19 days in a medium containing natural amino acids (light) and then switched to a medium containing heavy isotopically labeled arginine and lysine together with BIC or TTX or without a treatment (Ctrl). Upon the medium change, heavy amino acids were incorporated into nascent proteins, whereas the fraction of light pre-existing proteins decayed over time. The cells were harvested 1, 3 and 7 days after the medium change as well as just before the medium change ($t_0$). (**B**) Protein turnover rates are estimated using the fraction of remaining pre-existing protein signal (light [L]) relative to the total protein signal (heavy and light [H+L]). (**C**) Calculated protein half-lives in primary hippocampal cultures under basal neuronal activity (Ctrl, black), during BIC-induced homeostatic down-scaling (green) and during TTX-induced homeostatic up-scaling (violet). Protein half-lives for the different treatments are displayed in ascending order and the median half-life for each treatment is indicated. N = 3 biological replicates. Protein half-life distributions are significantly different for BIC- and TTX-treated neurons than for untreated controls (Mann-Whitney test, p<0.001). (**D**) Combinations of possible changes in protein synthesis and degradation rates and resulting changes in

*Figure 1 continued on next page*

*Figure 1 continued*

calculated half-lives. (**E**) To enable accurate and independent quantification of 'light' and 'heavy' signals, a fully 'semi-heavy'-labeled neuronal cell lysate was spiked into each sample to serve as an internal standard (IS). (**F**) Schematic MS1 spectrum for a dynamic SILAC sample with IS spike-in. For each condition and each time point, each peptide shows three signals, the light (L) pre-existing peptide, the heavy (H) new peptide and the semi-heavy (S) IS peptide. (**G**) Quantification of protein synthesis, degradation and abundance. Protein synthesis is quantified using the heavy (H) signal normalized to the semi-heavy (S) IS, whereas protein degradation is assessed using the light (L) signal of pre-existing proteins normalized to the IS and protein abundance is quantified using the sum of light and heavy signal normalized to the IS. (**H**) Combinations of possible changes in protein synthesis and degradation rates and resulting changes in protein turnover or protein abundance.

The online version of this article includes the following figure supplement(s) for figure 1:

**Figure supplement 1.** Heavy amino acid incorporation.
**Figure supplement 2.** Electrophysiological recordings.
**Figure supplement 3.** Precision of protein half-life determination.

---

proteins under all experimental conditions are presented in *Figure 1—figure supplement 3*. The corresponding protein half-lives range from several hours up to >14 days (*Figure 1C*). Under basal neuronal activity (Ctrl), a median protein half-life of 5.3 days was observed, which is consistent with previous studies from our lab (median half-life of 5.4 days) (*Dörrbaum et al., 2018*). Upon TTX-induced homeostatic up-scaling, the protein half-life distribution was significantly shifted towards longer half-lives (median half-life 5.7 days; Mann-Whitney test, p<0.001). An even stronger shift towards longer protein half-lives was observed upon BIC-induced down-scaling (median half-life 6.1 days; Mann-Whitney test, p<0.001), demonstrating that the neuronal proteome dynamics are affected by homeostatic scaling.

Which are the underlying mechanisms that bring about the observed changes in protein turnover? The most intuitive interpretation of the shift towards longer half-lives is that proteins become stabilized upon homeostatic scaling. However, the calculated protein half-life is influenced not only by changes in protein degradation but also by changes in protein synthesis (denominator in *Figure 1B*). Hence, altered protein half-lives can be a result of distinct combinations of altered synthesis and degradation rates (*Figure 1D*). To understand the protein dynamics that drive homeostatic scaling, we quantified independently the protein synthesis and degradation rates as well as the resulting changes in protein abundance and turnover. To enable precise quantification of protein synthesis (heavy channel) and degradation (light channel), we made use of an internal standard (IS), a fully labeled 'semi-heavy' neuronal lysate, that was spiked into all samples ('triple SILAC'; *Figure 1E*; see 'Materials and methods' section for details). The resulting MS1 spectra then contained three peaks per peptide: the light (L) signal of the pre-existing protein, the heavy (H) signal of the new protein and the semi-heavy (S) signal of the IS (*Figure 1F*). As the IS was present in all samples in equal amounts, the IS signal was used for normalization purposes to correct for experimental and MS variability (*Figure 1G*). This experimental design allowed us to quantify changes in protein synthesis and degradation independently. *Figure 1H* shows all possible combinations of changes in protein synthesis and degradation as well as their effect on protein abundance (up or down) and turnover (faster or slower turnover). The same color code is used throughout the manuscript. 'Warm' reddish colors indicate a resulting increase in protein abundance, whereas 'cold' blueish colors indicate a decrease in protein abundance.

## Homeostatic scaling is accomplished by substantial changes in protein synthesis and degradation

In total, we identified 7150 proteins throughout all experimental conditions. We first analyzed changes in protein synthesis and degradation over the entire duration of the down- and up-scaling experiments (1–7 days; see 'Materials and methods' section). After stringent filtering for dataset completeness, 3596 proteins were quantified for the down-scaling versus control comparison and 3609 proteins were quantified for the up-scaling versus control comparison (*Figure 2—source data 1*). Upon induction of homeostatic down-scaling (BIC treatment), a large fraction of the quantified proteome (43%), showed changes in protein synthesis and/or degradation (*Figure 2A*). Generally speaking, the regulation was biased toward decreased protein synthesis and/or degradation. Of the

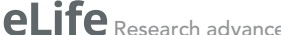

**Figure 2.** Regulation of protein synthesis and degradation rates during homeostatic scaling. (**A–C**) Regulation during BIC-induced homeostatic down-scaling. (**A**) Numbers of proteins with significantly altered ('regulated') synthesis and/or degradation rates. (**B**) Numbers of proteins that show the indicated types of regulation. (**C**) Behavior of individual proteins showing changes in protein synthesis (nascent proteins; x-axis) and degradation (pre-existing proteins; y-axis). Significantly regulated proteins (1% false discovery rate [FDR]) are displayed in color and proteins that are not significantly changed are displayed in light gray. Some individual proteins of interest are indicated by name. The inset shows the distribution of all quantified proteins with equal axis dimensions. (**D–F**) Regulation during TTX-induced homeostatic up-scaling. (**D**) Numbers of proteins with significantly altered ('regulated') synthesis and/or degradation rates. (**E**) Numbers of proteins that show the indicated types of regulation. (**F**) Behavior of individual proteins showing changes in protein synthesis (nascent proteins; x-axis) and degradation (pre-existing proteins; y-axis). Significantly regulated proteins (1% FDR) are displayed in color and proteins that are not significantly changed are displayed in light gray. Some individual proteins of interest are indicated by name. The inset shows the distribution of all quantified proteins with equal axis dimensions.

The online version of this article includes the following source data and figure supplement(s) for figure 2:

**Source data 1.** Protein regulation during homeostatic up- and down-scaling.

**Figure supplement 1.** Validation of selected proteins by western blot.

**Figure supplement 2.** Analysis of Caspase-3 stability.

regulated proteins, the largest group were proteins that exhibited decreased synthesis rates (703 proteins), followed by those exhibiting decreased degradation rates (496 proteins) and those exhibiting decreased protein turnover rates (less synthesis together with less degradation; 292 proteins). By contrast, only a few proteins showed increased synthesis rates (18 proteins), increased synthesis rates together with decreased degradation rates (nine proteins), increased degradation rates (11 proteins) or increased degradation rates together with reduced synthesis rates (13 proteins; *Figure 2B–C*). These results suggest that reduced protein degradation and reduced protein synthesis are the preferred cellular mechanisms to achieve changes in protein abundance during homeostatic down-scaling.

We next compared the changes observed during homeostatic down-scaling to the regulation produced by homeostatic up-scaling (TTX treatment). During up-scaling, 31% of the quantified proteins showed a change in synthesis and/or degradation (*Figure 2D*). Similar to the results for homeostatic down-scaling, following up-scaling, the largest groups of proteins were regulated by a decrease in synthesis (503 proteins), with the second-largest group being regulated by a decrease in degradation (332 proteins; *Figure 2E–F*). Again, comparably lower numbers of proteins showed increased synthesis rates (93 proteins), increased synthesis rates together with decreased degradation rates (80 proteins), increased degradation rates (42 proteins) or increased degradation rates together with reduced synthesis rates (59 proteins). This suggests that reduced protein degradation might be generally preferred over increased protein synthesis to achieve long-term proteomic up-regulation, and that reduced protein synthesis might be generally preferred over increased protein degradation to achieve long-term proteomic down-regulation.

To validate the observation that protein synthesis was reduced for large fractions of proteins during homeostatic scaling, we used a pulsed metabolic labeling technique that makes it possible to tag and quantify nascent proteins via western blot (*Dieterich et al., 2006*). Primary hippocampal cultures were treated with BIC, TTX or vehicle (Ctrl) for 24 hr, and nascent proteins were labeled with azidohomoalanine (AHA) during the last 2 hr of the treatment and linked to biotin via a click reaction (*Figure 3—figure supplement 1A*). Western blot analysis revealed that protein synthesis was

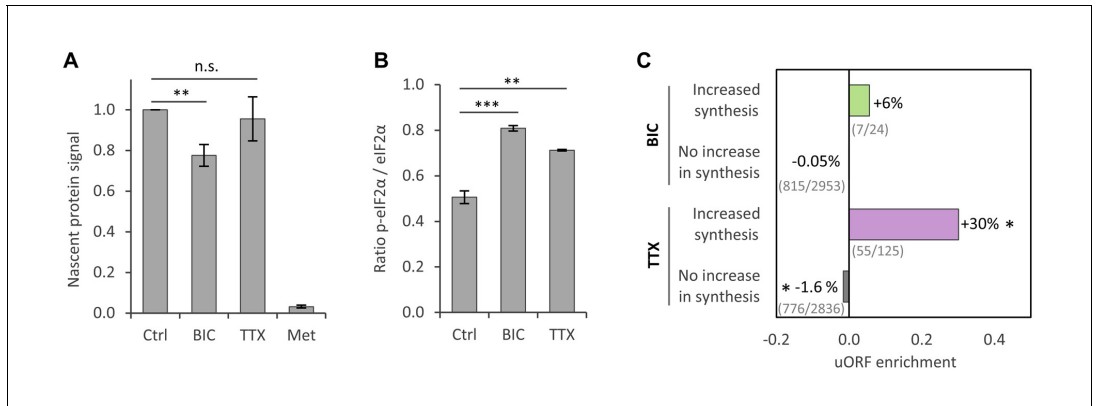

**Figure 3.** Regulation of protein synthesis during homeostatic scaling. (A) Quantification of the biotin-derived nascent protein signal via western blot. Error bars represent the standard error of the mean (SEM). N = 5 biological replicates. Protein synthesis was significantly reduced in BIC-treated hippocampal cultures compared to untreated controls (paired t-test, p<0.05), whereas the change in protein synthesis was not statistically significant for TTX-treated hippocampal cultures. (B) Quantification of phosphorylated as well as unmodified eIF2α via western blot. Error bars represent the SEM. N = 3 biological replicates. Phosphorylation of eIF2α at Ser51 was significantly increased after 24 hr of treatment with BIC (unpaired t-test, p-value<0.001) and TTX (unpaired t-test, p-value<0.01). (C) uORF enrichment analysis for the sets of proteins that do or do not show increased protein synthesis during BIC-induced down-scaling or TTX-induced up-scaling. uORF-containing genes are significantly over-represented in the group of proteins that showed increased synthesis during TTX treatment (hypergeometric test, p<0.05) and significantly under-represented in the group of proteins that did not show increased synthesis during TTX treatment (hypergeometric test, p<0.05). Numbers of uORF-containing transcripts per gene subset are shown in brackets. The rat genome was used as reference.

The online version of this article includes the following figure supplement(s) for figure 3:

**Figure supplement 1.** Quantification of nascent proteins via western blot.
**Figure supplement 2.** eIF2α phosphorylation.
**Figure supplement 3.** Autophagy and proteasomal degradation.

significantly reduced during BIC-induced down-scaling (*Figure 3A*; *Figure 3—figure supplement 1B*). Protein synthesis was also slightly decreased during TTX-induced up-scaling, although the difference did not reach statistical significance (*Figure 3A*; *Figure 3—figure supplement 1B*). These findings are consistent with our MS results after 1 day of treatment, in which a clear shift towards less protein synthesis was observed for BIC-treated cultures and a lesser effect was seen for TTX-treated cultures (see section 'Temporal regulation of protein synthesis and degradation during homeostatic scaling'). In addition, we validated the regulation of several individual proteins by western blot and found an excellent correlation between the SILAC-MS data and the western blot results (*Figure 2—figure supplement 1*).

To exclude the possibility that the observed changes in protein synthesis and degradation were a result of cellular stress, all neuronal preparations were visually inspected under the microscope prior to harvest and no signs of stress or apoptosis (blebbing or fragmenting in the dendrites) were observed. In addition, we searched for regulation of the apoptotic marker Caspase-3 (Casp-3) in the MS data, as the degradation rate of Casp-3 is a measure of ongoing apoptosis (*Tawa et al., 2004*). No increase in the degradation of Casp-3 was observed in BIC- and TTX-treated hippocampal cultures in comparison to untreated controls (*Figure 2—figure supplement 2*), indicating that the treatment did not impair cell viability.

## Reduced protein synthesis is associated with eIF2α phosphorylation, whereas proteasomal degradation and autophagy are unaffected during homeostatic scaling

We next asked whether the observed decrease in protein synthesis was the result of a targeted regulation of large numbers of individual proteins or whether translation was altered using general translation control mechanisms. Translation initiation is catalyzed by multiple initiation factors and is a major target of translational control. For example, phosphorylation of the eukaryotic initiation factor two alpha (eIF2α) at Ser51 inhibits translation (*Dever, 2002*; *Hinnebusch, 2005*). Western blot analysis revealed a significant increase in the phosphorylation of eIF2α at Ser51 during TTX-induced up-scaling (40% increase) and an even stronger increase during BIC-induced down-scaling (60% increase; *Figure 3B*; *Figure 3—figure supplement 2*). The changes in Ser51 phosphorylation correlate well with the decrease in protein synthesis observed by SILAC-MS and western blot analysis. It is likely, however, that the synthesis of individual proteins, as well as the process of translation, is additionally regulated by other mechanisms during homeostatic scaling.

Despite a general inhibition of translation initiation, phosphorylation of eIF2α also leads to a paradoxical increase in the translation of mRNAs that contain upstream open reading frames (uORFs) in their 5′-UTRs (*Dever, 2002*; *Hinnebusch, 2005*). Consistent with this, a subset of proteins showed increased synthesis during homeostatic up-scaling (*Figure 2E–F*) and down-scaling (*Figure 2B–C*). We analyzed these proteins in more depth and found a significant over-representation of genes containing uORFs within the genes encoding the group of proteins that showed increased synthesis during TTX-induced up-scaling. In addition, we found a slight under-representation of genes containing uORFs in genes encoding the group of proteins that did not show increased synthesis during up-scaling. For BIC-induced down-scaling, we also found an over-representation of genes containing uORFs within those encoding the group of proteins that showed increased synthesis, though this was not statistically significant (*Figure 3C*).

Although decreased degradation rates were also observed for large numbers of proteins during homeostatic up- and down-scaling (*Figure 2B–F*), we did not detect changes in the activity of proteasomes (*Figure 3—figure supplement 3A*) or differences in ongoing autophagy (*Figure 3—figure supplement 3*; *Figure 3—figure supplement 3B*).

## Functional analysis of regulated proteins

What are the functions of the proteins that are regulated by homeostatic scaling? GO over-representation analysis revealed that distinct functional groups were over-represented within the subsets of proteins that have certain regulation patterns (*Supplementary file 1*). During down-scaling, we found an over-representation of secretory vesicle and secretory granule proteins, as well as extracellular proteins that were up-regulated by increased synthesis. The strongest increase in synthesis was observed for Secretogranin-2 (Sgc2), a secretory protein that is released into the synaptic cleft in a

calcium-dependent manner (*Fischer-Colbrie et al., 1995*) and is implicated in several neurological diseases (*Agneter et al., 1995*; *Kandlhofer et al., 2000*; *Kaufmann et al., 1998*; *Lechner et al., 2004*; *Mahata et al., 1992*; *Medhurst et al., 2001*). Among the proteins that were up-regulated by decreased degradation, we found an over-representation of proteins involved in translation (such as ribosomal proteins and proteins associated with the aminoacyl-tRNA synthetase multienzyme complex), as well as proteins involved in proteolysis (especially proteins of the base and lid of the proteasome regulatory particle). On the other hand, the group of proteins that were down-regulated by reduced synthesis showed an over-representation of synaptic proteins, which were associated with the GO terms 'excitatory synapse', 'postsynaptic density', 'presynaptic active zone', 'chemical synaptic transmission', 'regulation of synaptic plasticity' and others. In addition, there was an over-representation of axonal and dendritic proteins and of proteins that are located in the neuronal cell body. Among the 5% of proteins that showed the strongest reduction in protein synthesis were known regulators of synaptic plasticity, such as Camk2a, PSD-95, Neuronal pentraxin-1 (Nptx1) and the glutamate receptor subunits GluA1, GluA3 and GluN2b. For proteins that were down-regulated by both decreased synthesis and increased degradation, a massive over-representation of synaptic and dendritic proteins was found, especially for the terms 'excitatory synapse', 'postsynaptic density' and 'dendritic shaft'. In this group, the strongest regulation was found for synaptopodin (Synpo).

During homeostatic up-scaling, the group of proteins that was up-regulated by both increased synthesis and decreased degradation showed an over-representation of secretory granule proteins and of proteins involved in signaling, cell communication and stimulus response. The proteins with the strongest regulation include secretogranin-1, 3 and 5 (Scg1, Scg3, and Scg5), neuronal pentraxin-1 (Nptx1), neuronal pentraxin receptor (Nptxr), ProSAAS, Carboxypeptidase E (Cpe) and Homer3. For the group of proteins that were up-regulated by decreased degradation, several GO terms were over-represented, including 'magnesium ion binding', 'nucleoside diphosphate phosphorylation', 'carbohydrate metabolic process', 'intracellular signal transduction' and 'protein modification process'. The protein with the strongest decrease in degradation was the Neuronal calcium sensor 1 (Ncs1), a protein that was previously described in the context of synaptic plasticity (*Weiss et al., 2010*). Among the proteins that were down-regulated by decreased synthesis, we found an over-representation of many synaptic terms, including 'GABA-ergic synapse' and 'glutamatergic synapse'. The regulation of synaptic proteins is described in more detail below. Within the group of proteins that were down-regulated by increased degradation, there was an over-representation of proteins involved in mRNA translation. Note that for proteins involved in translation, the opposite regulation (decreased degradation) was observed during BIC-induced down-scaling. Proteins that were down-regulated by both decreased synthesis and increased degradation showed an over-representation of endoplasmic reticulum (ER) proteins and of proteins of the eukaryotic translation initiation factor 3 complex.

## The synaptic proteome is substantially remodeled during homeostatic scaling

Our dataset contains numerous synaptic proteins and we were able to quantify the behavior of 301 and 302 synaptic proteins during homeostatic down-scaling (*Figure 4*) and up-scaling (*Figure 5*), respectively, of which large fractions exhibited regulation. During BIC-induced down-scaling, 59% of the synaptic proteins showed a change in synthesis and/or degradation (*Figure 4—figure supplement 1A*). During TTX-induced up-scaling, 52% of the synaptic proteins showed a change in synthesis and/or degradation (*Figure 4—figure supplement 1B*).

Consistent with the concept of synaptic down-scaling, most synaptic proteins were down-regulated upon BIC treatment. For most of these proteins, the mechanism of down-regulation is a decrease in protein synthesis. Prominent examples in this group of proteins were the AMPAR subunits GluA1 and GluA3, fitting with the observation that AMPAR surface expression is decreased during homeostatic down-scaling (*Turrigiano, 2012*; *Turrigiano, 2008*). Interestingly, no significant change in degradation was observed for GluA1 and GluA3. Under basal neuronal activity, the half-lives of GluA1 and GluA3 are 2.8 days and 3.5 days, respectively. These results indicate that long-lasting changes of AMPAR surface expression (in the range of days) during homeostatic down-scaling are probably accomplished by continuous degradation of pre-existing copies at an unchanged degradation rate, together with a decreased supply of newly synthesized copies.

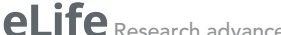

**Figure 4.** Regulation of synaptic proteins during BIC-induced homeostatic down-scaling. Down-scaling regulated proteins associated with excitatory glutamatergic synapses (**A**) or inhibitory GABAergic synapses (**B**). Changes in protein synthesis and/or degradation are indicated by a color code. 'Warm' reddish colors indicate a change in synthesis/degradation that leads to increased protein copy numbers and 'cold' blueish colors indicate a change that leads to decreased protein copy numbers. Source data are provided in **Figure 4—source data 1**. Synaptic proteins that are also regulated by TTX (see **Figure 5**) are marked with an asterisk.

The online version of this article includes the following source data and figure supplement(s) for figure 4:

**Source data 1.** Synaptic protein regulation during homeostatic down-scaling.
**Figure supplement 1.** Regulation of synaptic proteins during homeostatic scaling.

The same pattern of regulation (decreased synthesis and unchanged degradation) was also observed for the AMPAR auxiliary subunit TARP-γ8, other glutamate receptor subunits (GluK2, GluN2B and mGluR5), scaffold proteins of the postsynaptic density (such as PSD-95, Homer3, SAP-102, DAP-3 and DAP-4), cell adhesion molecules (such as Neuroligins and Neurexins), as well as proteins involved in axon guidance/growth, the synaptic vesicle cycle and other synaptic functions. Only

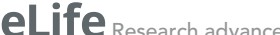

**Figure 5.** Regulation of synaptic proteins during TTX-induced homeostatic up-scaling. Up-scaling regulated proteins associated with excitatory glutamatergic synapses (**A**) or inhibitory GABAergic synapses (**B**). Changes in protein synthesis and/or degradation are indicated by a color code. 'Warm' reddish colors indicate a change in synthesis/degradation that leads to increased protein copy numbers and 'cold' blueish colors indicate a change that leads to decreased protein copy numbers. Source data are provided in *Figure 5—source data 1*. Synaptic proteins that are also regulated by BIC (see *Figure 4*) are marked with an asterisk.

The online version of this article includes the following source data for figure 5:

**Source data 1.** Synaptic protein regulation during homeostatic up-scaling.

four proteins were down-regulated by an exclusive increase in protein degradation (Syn3, Sept6, rGVAT and Epb41l3), and an additional seven proteins were down-regulated by both decreased synthesis and increased degradation (Akap5, Atp2b2, Cacna1b, Homer1, Rgs12, Sptbn2 and Synpo). Only a few synaptic proteins were up-regulated during homeostatic down-scaling. Three proteins were up-regulated by increased synthesis (Itgb1, Ptprn and Slc6a17), 17 proteins were up-regulated

by decreased degradation and two proteins were up-regulated by both increased synthesis and decreased degradation (Bdnf and Pcdh8).

The majority of the synaptic proteins that were affected by BIC-induced down-scaling were also regulated during TTX-induced up-scaling and vice versa (111 proteins; *Figure 4—figure supplement 1C*). In contrast to BIC-induced down-scaling, during TTX-induced up-scaling, the number of synaptic proteins that were up-regulated (66 proteins) was similar to the number that was down-regulated (79 proteins) (*Figure 5*).

Consistent with the increased AMPAR surface expression observed during homeostatic up-scaling (*Turrigiano, 2012*; *Turrigiano, 2008*), increased protein synthesis was observed for GluA1, but not for GluA2 and GluA3. In addition to GluA1, ten additional synaptic proteins were up-regulated by increased synthesis only (Cntn2, Epha4, Frrs1l, NLG-3, Noelin, Prrt2, Rapgef2, Syt7, vGlut2 and Vamp7). Note that seven of these ten proteins were also regulated during homeostatic down-scaling and showed opposite changes (less synthesis). A comparably large group of 44 synaptic proteins was up-regulated by decreased degradation. This group includes mGluR5, Camk2b/g, PKC as well as scaffold proteins of the postsynaptic density (such as PSD-95, DAP1, DAP3 and Shank1) and proteins that are involved in the synaptic vesicle cycle (such as Syn1, Syngr1, and Stx1a). In addition, 11 proteins were up-regulated by both increased synthesis and decreased degradation. These proteins include the AMPAR regulator Camk2a, the postsynaptic density scaffold protein Homer3, Nitric oxide synthase 1 (Nos1) and presynaptic proteins that are involved in the synaptic vesicle cycle, such as Syt5, vGlut1 and SV2b. Down-regulation of synaptic proteins was mainly accomplished by a decrease in protein synthesis. 76 synaptic proteins exhibited decreased synthesis rates, including the AMPAR subunits GluA2/3, postsynaptic density scaffold proteins (such as Homer1, SAP-102 and DAP-4), cell adhesion molecules, and proteins involved in axon guidance/growth or the synaptic vesicle cycle. Three synaptic proteins were down-regulated by decreased synthesis as well as increased degradation (Nrxn1, Sipa1l1 and Atp1a2), while no synaptic proteins were found to be down-regulated during homeostatic up-scaling by increased degradation alone.

## Regulation of protein synthesis and degradation elicit changes in protein abundance and turnover

Simultaneous changes in protein synthesis and degradation can elicit changes in either protein abundance or turnover (*Figure 1H*). Changes in protein abundance are influenced by the directionality and strength of the changes in synthesis and degradation, as well as by the initial protein abundance. We next analyzed the changes in protein abundance as described in *Figure 1G*. Upon BIC-induced down-scaling, 208 proteins were significantly decreased in abundance and 185 were significantly increased in abundance (*Figure 6A*). Although the number of analyzed proteins was similar, the number of significantly regulated proteins was considerably lower when analyzing changes in relative protein abundance compared to changes in the nascent proteome (synthesis) and pre-existing proteome (degradation) separately. There are two reasons for this discrepancy. First, many proteins showed decreased synthesis as well as decreased degradation, leading to a slower protein turnover (*Figure 2B*), while protein abundance remains mostly unaffected. Second, the independent analysis of the nascent or pre-existing proteome is more sensitive to small changes in protein synthesis or degradation, respectively, compared to the analysis of the entire proteome. Upon TTX-induced up-scaling, 459 proteins were significantly increased and 493 proteins were significantly decreased in abundance (*Figure 6B*).

## Common and unique protein regulation during homeostatic up- and down-scaling

A comparison of the regulation observed upon both treatments revealed that distinct sets of proteins showed significant changes in abundance exclusively upon BIC-induced down-scaling (197 proteins) or TTX-induced up-scaling (735 proteins; *Figure 6C*). This observation suggests that, at least in part, different proteins are responsible for the phenotypes of homeostatic up- and down-scaling. However, this comparison also revealed that significantly more proteins showed changes in abundance during both treatments (194 proteins) than expected by chance (Fisher's exact test), suggesting that partially overlapping sets of regulated proteins were responsible for up- and down-scaling. We analyzed the changes observed for the proteins that were affected by both treatments in more

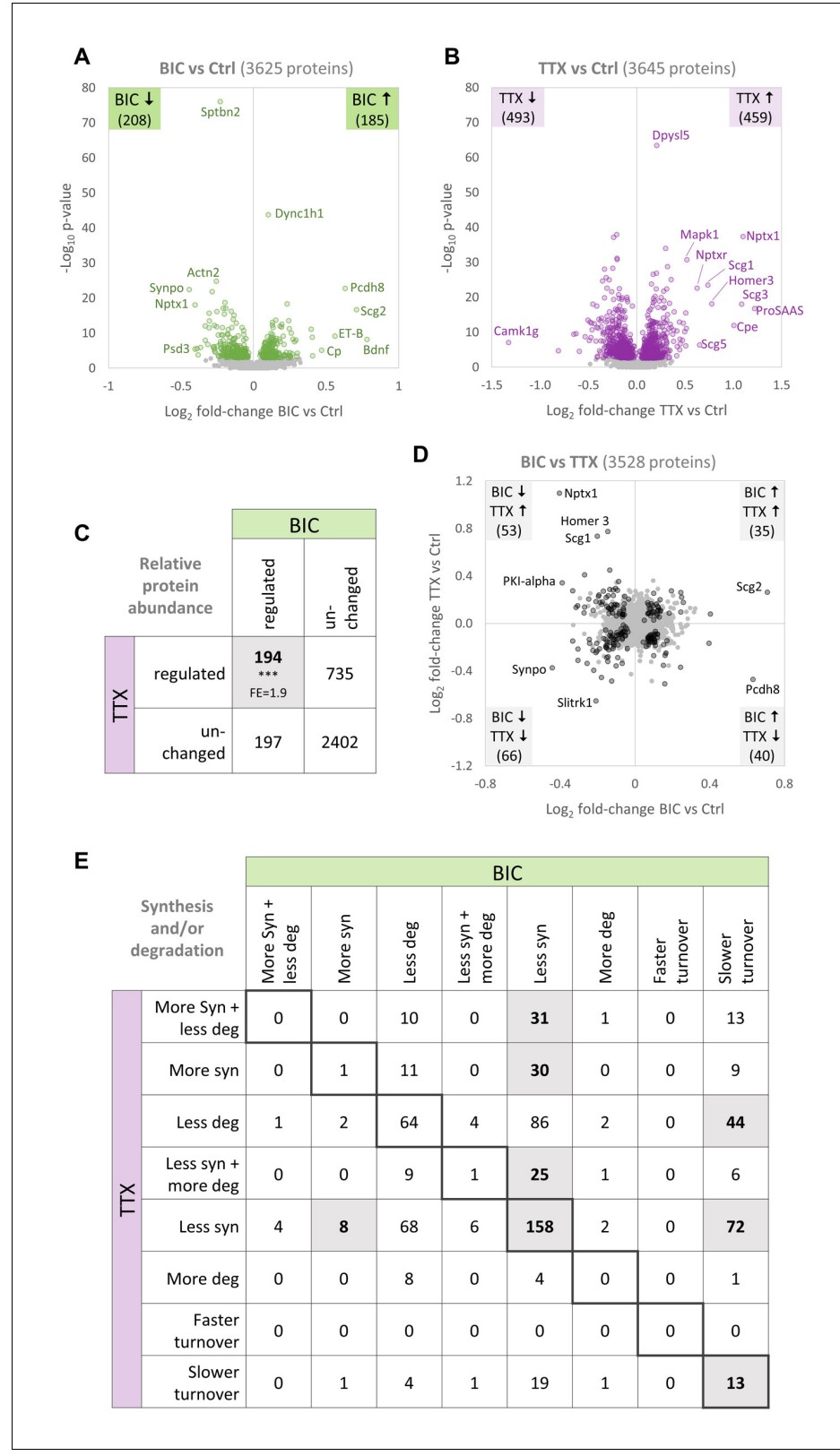

**Figure 6.** Comparison of protein changes during homeostatic up-scaling and down-scaling. (**A–B**) Differential regulation of protein abundance during (**A**) BIC-induced down-scaling or (**B**) TTX-induced up-scaling compared to untreated controls. Significantly regulated proteins (FDR <0.01) are shown in color. Proteins that were not significantly regulated are shown in gray. (**C**) Numbers of proteins that were differentially regulated or unchanged

*Figure 6 continued on next page*

*Figure 6 continued*

in abundance during BIC-induced down-scaling and TTX-induced up-scaling. Significantly more proteins were regulated by both treatments than expected by chance (Fisher's exact test, ***p<0.001). Fold-enrichment (FE) is indicated. (D) Fold-changes in relative protein abundance during TTX-induced up-scaling (y-axis) versus BIC-induced down-scaling (x-axis). Proteins that are significantly regulated by both treatments (BIC and TTX) are shown in black and proteins that are not significantly regulated are shown in gray. Source data are provided in *Figure 6—source data 1*. (E) Detailed comparison of changes in protein synthesis and/or degradation. Significantly over-represented combinations are highlighted. P-values and fold-enrichment are given in *Figure 6— figure supplement 1B*.

The online version of this article includes the following source data and figure supplement(s) for figure 6:

**Source data 1.** Changes in protein abundance during homeostatic up- and down-scaling.
**Figure supplement 1.** Comparison of changes in protein synthesis and/or degradation during homeostatic up- and down-scaling.

detail (*Figure 6D*), and found that similar numbers of proteins showed the same regulation or opposite regulation. Proteins that show the same regulation upon both treatments might be 'general plasticity proteins' that are involved in sensing an offset from basal activity levels, regardless of the sign of that offset. 35 proteins (e.g. Scg2) were up-regulated during up-scaling and down-scaling, and 66 proteins (e.g. Synpo) were down-regulated during both treatments. By contrast, proteins that show opposite regulation upon the opposite treatments might be 'polarity proteins' that sense the polarity of the offset or determine the polarity of the compensatory scaling. 53 proteins were up-regulated upon TTX-induced up-scaling and down-regulated upon BIC-induced down-scaling (e.g. Nptx1, Homer1 and Scg1) and 40 proteins were up-regulated during BIC-induced down-scaling and down-regulated during TTX-induced up-scaling (e.g. Pcdh8, Pcdh17, Vgf and several ribosomal proteins).

We analyzed the underlying types of regulation in more detail and found that several combinations of regulation upon BIC and TTX treatment were significantly over-represented (*Figure 6E*, *Figure 6—figure supplement 1*). Among these patterns, we found an over-representation of proteins that showed decreased synthesis upon both treatments as well as proteins that showed decreased synthesis upon BIC treatment and decreased synthesis together with increased degradation upon TTX treatment. Proteins of these groups might be affected by molecular pathways that are involved in both homeostatic up- and down-scaling, such as the global decrease in translation caused by eIF2α phosphorylation. Only one protein showed an increase in synthesis upon both treatments. By contrast, many proteins that showed increased synthesis upon one treatment, showed the opposite regulation (decreased synthesis) upon the other treatment. Proteins of these groups might be part of the molecular pathways that underlie homeostatic up- and down-scaling, and changes in their synthesis rates (increased or decreased synthesis) might define the polarity of the scaling. The fact that most over-represented combinations involve a change in protein synthesis for both treatments suggests that changes in the nascent proteome mainly drive homeostatic scaling and determine the polarity of the scaling, consistent with the fact that protein synthesis is required for homeostatic scaling (*Schanzenbächer et al., 2016*).

## Temporal regulation of protein synthesis and degradation during homeostatic scaling

The analysis of the data obtained over the entire time course of the experiment (1, 3 and 7 days treatment) enables the sensitive detection of the long-scale regulation of protein synthesis or degradation, but does not consider shorter-term temporal changes in the strength or direction of the regulation. To inspect the temporal dynamics of protein synthesis and degradation, we analyzed the individual time points separately and found that many proteins exhibited temporal changes in the strength and direction of regulation during homeostatic scaling. After 1 day of BIC treatment, more than 600 proteins exhibited a change in synthesis and, on average, protein synthesis rates were reduced by approximately 20% compared to untreated controls (*Figure 7A*). This is in good agreement with the decrease in protein synthesis detected by AHA labeling and western blot (*Figure 3A*). While the average fold-change (*Figure 7A*) and the average regulation strength (*Figure 7B*) of protein synthesis decreased over time, the opposite trend was observed for protein

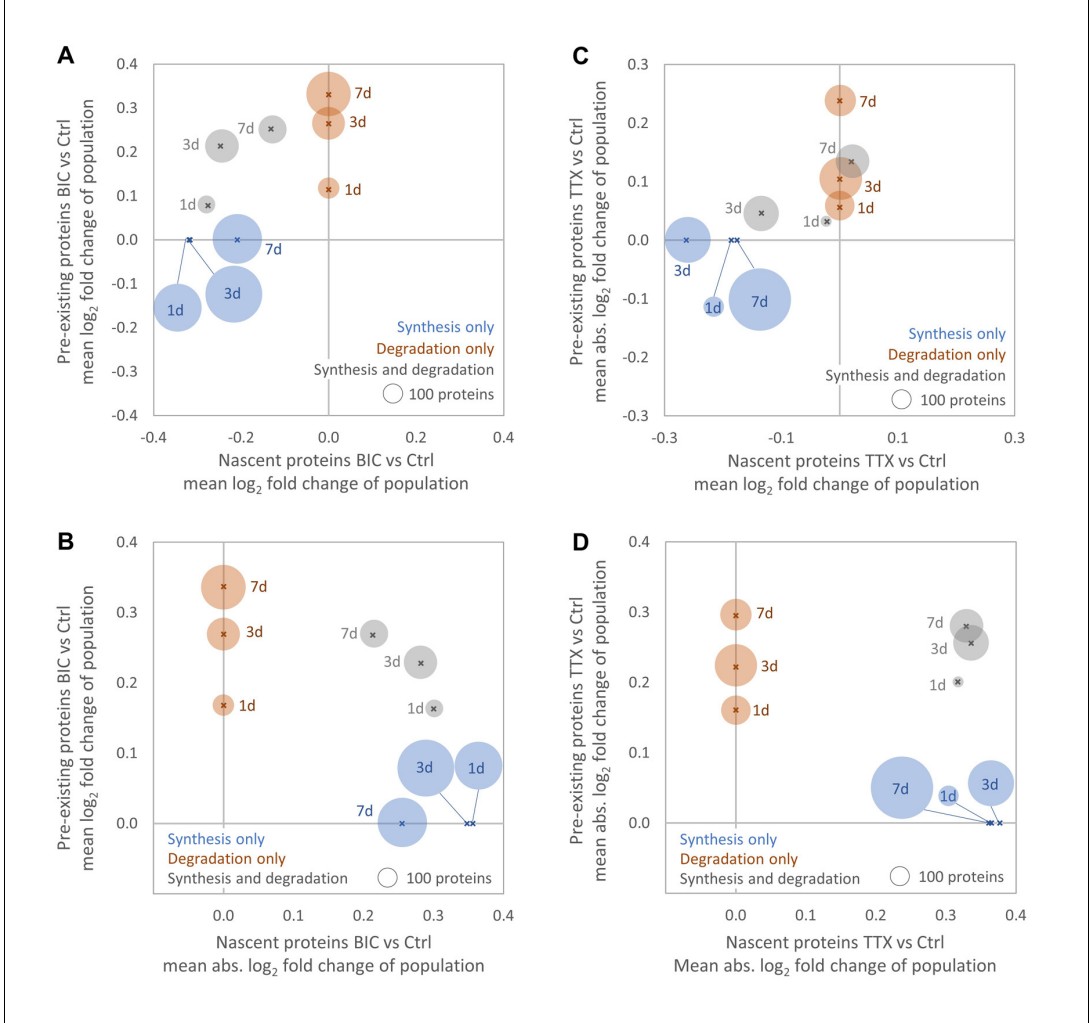

**Figure 7.** Temporal changes in protein synthesis and degradation during homeostatic scaling. Global changes in protein synthesis and/or degradation after 1, 3 and 7 days of BIC-induced down-scaling (**A, B**) or TTX-induced up-scaling (**C, D**). For each time point, the average fold-change (**A, C**) and the average strength of regulation (**B, D**) of all proteins that are significantly regulated by synthesis (blue), degradation (brown) or both (grey) are shown. The circle size is proportional to the number of regulated proteins.

degradation. During TTX-induced up-scaling, the numbers of proteins with significantly regulated synthesis rates increased over time (*Figure 7C*). These protein sub-populations showed, on average, a decrease in protein synthesis with the strongest average decrease observed after 3 days of treatment. The average regulation strength was similar for all time points (*Figure 7D*). The sub-populations of proteins with significantly altered degradation rates showed, on average, a decrease in degradation for all time points. Both the average fold change (*Figure 7C*) and the average regulation strength (*Figure 7D*) increased over time. Note that the average change in protein synthesis after 1 day of TTX treatment was consistent with the result obtained by AHA labeling and western blot analysis (*Figure 3A*). The sub-population of proteins with significant changes in synthesis showed, on average, a decrease in synthesis. However, as this sub-population was small compared to the total proteome (115 proteins regulated by synthesis only and 33 proteins regulated by synthesis and degradation), the changes were not detectable by western blot.

We next analyzed the temporal changes in synthesis and degradation for individual proteins. To identify groups of proteins that have similar behavior during homeostatic scaling, all quantified proteins were clustered using their temporal profiles of pre-existing and newly synthesized proteins (kmeans clustering), and the resulting clusters were manually assigned to upper-level groups (*Figure 8*, *Figure 8—figure supplements 1,2*, *Figure 8—source data 1*). During down-scaling, there

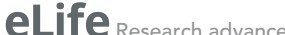

**Figure 8.** Temporal profiles of protein synthesis and degradation during homeostatic scaling. Predominant temporal profiles of pre-existing and newly synthesized proteins observed during BIC-induced down-scaling and TTX-induced up-scaling. Proteins were clustered using their temporal profiles of pre-existing and newly synthesized proteins (k-means clustering), and the resulting clusters of proteins with similar profiles were manually assigned to upper-level groups (see *Figure 8—figure supplements 1,2*). (**A**) Description of graphs in panels (**B–D**) showing exemplary profiles. (**B–D**) The bar diagrams show the numbers of individual proteins assigned to upper-level groups that are mainly characterized by changes in synthesis (**B**), degradation (**C**) or both (**D**). Representative temporal profiles of proteins regulated during BIC-induced down-scaling (green) or TTX-induced up-scaling (purple) are shown.

The online version of this article includes the following source data and figure supplement(s) for figure 8:

**Source data 1.** Temporal protein regulation during homeostatic up- and down-scaling.
**Figure supplement 1.** Clusters of temporal protein profiles observed during BIC-induced down-scaling.
**Figure supplement 2.** Clusters of temporal protein profiles observed during TTX-induced up-scaling.

were several clusters of proteins (containing 967 proteins) that were mainly regulated by changes in protein synthesis, either over the entire time course ('persistent') or at distinct time points (early, intermediate or late). 192 proteins belong to clusters that showed a persistent decrease in protein synthesis (*Figure 8—figure supplement 1*). These clusters contain many synaptic proteins, such as the postsynaptic scaffold protein PSD-95 (*Figure 8B*). 184 proteins belong to clusters that showed a transient decrease in protein synthesis after 1 day of BIC treatment ('early peak'), such as the candidate plasticity gene 2 (Cpg2), which is involved in activity-dependent AMPAR internalization (*Loebrich et al., 2013*). The largest number of proteins was assigned to clusters that showed a transient change in protein synthesis after 3 days of BIC treatment (337 proteins; 'intermediate peak'), such as the postsynaptic density protein Disks large-associated protein 2 (DAP-2). Comparably smaller numbers of proteins were assigned to clusters with transient changes in protein synthesis after 3 and 7 days (209 proteins; 'intermediate and late peak') or after 7 days of BIC treatment (89 proteins; 'late peak'). In addition, 105 proteins were assigned to clusters that showed a decreasing regulation strength over time. A representative protein of this group is Bdnf, which showed the strongest increase in synthesis after 1 day of BIC treatment and decreasing regulation strength after 3 and 7 days of treatment. By contrast, for BIC treatment, there was no cluster of proteins that showed an increasing strength of changes in protein synthesis over time.

When TTX-induced up-scaling was compared to BIC-induced down-scaling, different major temporal profiles were observed for protein synthesis. During up-scaling, no clusters with persistent changes in synthesis over the entire time course or with transient changes in synthesis after 1 day of treatment were observed. Instead, more proteins were assigned to clusters that showed transient changes in synthesis after 3 days (231 proteins; 'intermediate peak'), 3 days and 7 days (287 proteins; 'intermediate and late peak') or 7 days (294 proteins; 'late peak') of treatment. In contrast to BIC-induced down-scaling, 135 proteins showed an increasing regulation strength over time, whereas only 20 proteins showed a decreasing regulation strength. For both down-scaling and up-scaling, there were several clusters of proteins that were mainly regulated by changes in degradation (*Figure 8C*). For both treatments, most proteins were assigned to clusters with transient changes in degradation after 3 days of treatment (150 proteins for BIC and 269 proteins for TTX). Comparably lower numbers of proteins were assigned to clusters with transient changes in degradation after 7 days of treatment (125 proteins for BIC and 179 proteins for TTX). In addition, several clusters showed increasing regulation strength over time (98 proteins for BIC and 84 proteins for TTX), whereas there were no clusters with decreasing regulation strength for both treatments.

Proteins that were regulated by both synthesis and degradation were also clustered and assigned to upper-level groups using the timing and the polarity (up- or down-regulation of the nascent and pre-existing proteins) of the observed regulation (*Figure 8D*). For BIC-induced down-scaling, only a few proteins were assigned to clusters that showed changes in synthesis and degradation leading to the same effect (15 proteins; nascent and pre-existing proteins both up-regulated or both down-regulated), whereas a larger number of proteins were assigned to clusters with changes in synthesis and degradation leading to the opposite effect (72 proteins). A representative example of this group is Prkce, which showed a down-regulation of nascent proteins (less synthesis) and an up-regulation of pre-existing proteins (less degradation). The opposite trend was observed for TTX-induced up-scaling. 242 proteins were assigned to clusters with changes in synthesis and degradation leading to the same effect. A representative example for this group is the voltage-dependent L-type calcium channel subunit Cacnb1, which showed up-regulation of both nascent proteins (more synthesis) and pre-existing proteins (less degradation). Only 40 proteins were assigned to clusters that showed changes in synthesis and degradation leading to the opposite effect.

Simultaneous regulation of protein synthesis and degradation that lead to the same protein abundance outcome, which is predominantly seen for TTX-induced up-scaling, can elicit a stronger change in relative protein abundance, whereas regulation of protein synthesis and degradation that lead to the opposite abundance outcomes, as predominantly seen for BIC-induced down-scaling, mainly influence protein turnover. In addition to the direction of the regulation, we also observed differences in the timing of the regulation of proteins synthesis and degradation during BIC and TTX treatments. For BIC-induced down-scaling, several clusters (containing 197 proteins) showed first a change in synthesis followed by a delayed change in degradation. A representative example for this group is the synaptic protein Teneurin-2, which showed decreased synthesis after 1 and 3 days of treatment followed by decreased degradation after 3 and 7 days of treatment. No protein cluster

with this predominant behavior was observed for TTX-induced up-scaling. By contrast, the opposite trend was observed for TTX. Two clusters (containing 67 proteins) showed first a change in degradation followed by a delayed change in synthesis, as shown for the representative protein Cacnb3. On the other hand, there was no protein cluster with this behavior under BIC treatment.

## Discussion

Here, we used dynamic SILAC labeling and MS to quantify changes in protein synthesis, degradation, abundance and turnover during homeostatic scaling in cultured neurons. We found that the neuronal proteome was massively remodeled during both down-scaling and up-scaling. Synaptic proteins were especially affected by the scaling. More than half of the quantified synaptic proteins were regulated. Most proteins were regulated by a decrease in synthesis or degradation. A small and mostly treatment-specific fraction of proteins showed increased synthesis rates.

In general, a cell has different options to adjust the abundance of its proteins. For protein up-regulation, the cell can produce more protein (more synthesis) or stabilize existing copies (less degradation) or do both. For protein down-regulation, a cell can reduce the synthesis or enhance the degradation or both. Our data demonstrate that decreased protein degradation and decreased protein synthesis are the preferred cellular mechanisms to achieve increased or decreased protein abundance, respectively, during homeostatic scaling. This preference was observed for both homeostatic up- and down-scaling, suggesting that decreasing protein synthesis or degradation might be a generally preferred mechanism to accomplish long-term changes in the neuronal proteome. Reduced protein synthesis or degradation rates might be preferred over increased degradation or synthesis rates, respectively, because it is the more energy-saving way to achieve proteomic changes. The signal processing power, and hence the cognitive capability, of the brain is limited by the availability of energy. Most of the brain's energy budget is required for signaling-related processes, and reduced 'housekeeping costs' (including costs associated with protein synthesis and degradation), provide a cognitive advantage (*Engl and Attwell, 2015*).

Mechanistically spoken, the decrease in protein synthesis observed during homeostatic scaling can be explained, at least in part, by eIF2α phosphorylation (*Figure 3B*), which inhibits translation initiation but also increases the translation of mRNAs that contain uORFs in their 5''UTRs. Phosphorylation of eIF2α was previously identified as an underlying mechanism of long-term depression induced by metabotropic glutamate receptor activation (mGluR-LTD), another protein synthesis-dependent form of synaptic plasticity (*Di Prisco et al., 2014*).

The proteomic response to up- and down-scaling exhibited both common and unique features. Many proteins were exclusively affected by TTX treatment (735 proteins) or BIC treatment (197 proteins, *Figure 6C*). In addition, a significantly over-represented fraction of proteins (194 proteins) showed a change in abundance upon both treatments (*Figure 6C*). Within the group of proteins that exhibited changes for both up- and down-scaling, some proteins showed the same type of regulation (increased or decreased abundance) upon both treatments (101 proteins), whereas an equal number of proteins exhibited opposite regulation upon BIC or TTX treatment (93 proteins; *Figure 6D*). How can the same underlying mechanism (e.g. eIF2α phosphorylation) result in different phenotypes? Although there is an over-representation of proteins that showed decreased synthesis upon both treatments, the proteins that showed increased synthesis were mostly exclusive to either BIC-induced down-scaling or TTX-induced up-scaling (*Figure 6E*), leading to distinct proteomic changes during up- and down-scaling. This specificity might be achieved by transcriptional regulation.

The nascent proteome is regulated by the mRNAs that are available for translation, especially those that contain uORFs and are preferentially translated upon eIF2α phosphorylation. Consistently, bidirectional transcriptional regulation has been observed in previous studies after 6 hr of BIC or TTX treatment in primary cortical cultures (*Schaukowitch et al., 2017*). Several of the proteins that showed increased synthesis during BIC-induced down-scaling in our data were previously reported to be up-regulated at the transcriptional level. For example, 27 proteins showed increased synthesis during homeostatic down-scaling in our data. Transcripts for 11 of these proteins were previously quantified during homeostatic down-scaling induced by 2 days of picrotoxin treatment (*Rajman et al., 2017*) or 6 hr of BIC treatment (*Schaukowitch et al., 2017*), of which 10/11 were significantly up-regulated. Also, transcriptional up-regulation was previously reported

(*Schaukowitch et al., 2017*) for 7 out of 8 proteins that showed increased synthesis during TTX-induced up-scaling.

Further specificity might be obtained by local changes in eIF2α phosphorylation. Upon BIC or TTX treatment, eIF2α might be phosphorylated in distinct neuronal cell types and/or at distinct sub-cellular localizations to repress general translation locally and to enhance the translation of locally enriched uORF-containing transcripts. In addition, the synthesis rates of individual proteins as well as translation in general might be regulated by other mechanisms. Previous studies demonstrated that post-transcriptional regulation by micro-RNAs, which regulate the pool of 'translatable' mRNAs, are critically involved in homeostatic up-scaling (*Cohen et al., 2011*; *Fiore et al., 2014*) as well as in down-scaling (*Hou et al., 2015*; *Letellier et al., 2014*; *Rajman et al., 2017*). Besides transcriptional and translational regulation, proteomic changes are also accomplished by changes in protein degradation. Despite the general decrease in protein degradation observed in our data, no change in proteasome activity or ongoing autophagy was observed, suggesting that protein degradation is regulated at the level of ubiquitin ligation by specific E3 ligases. During homeostatic up- and down-scaling, distinct sets of proteins showed altered degradation rates and there was no over-representation of proteins that were similarly affected by both treatments.

Synaptic proteins were especially affected by homeostatic up- and down-scaling. Of note, we observed extensive changes in both postsynaptic and presynaptic proteins, suggesting that both sides of the synapse are modified during homeostatic scaling. It is generally agreed that homeostatic scaling at mammalian synapses uses postsynaptic mechanisms that evoke changes in AMPAR surface expression. Consistent with this notion, we observed a decreased synthesis of the AMPAR subunits GluA1 and GluA3 and of the auxiliary subunit TARP-γ8 during homeostatic down-scaling, as well as increased synthesis of GluA1 during homeostatic up-scaling. Presynaptic mechanisms are less explored, but also contribute to homeostatic scaling (*Davis and Müller, 2015*; *Murthy et al., 2001*; *Turrigiano, 2012*; *Vitureira et al., 2012*). For instance, previous studies have described enhanced glutamate transporter (vGluT) expression in pre-synaptic terminals upon activity blockade (*De Gois et al., 2005*; *Erickson et al., 2006*), which might enable the neurons to package more neurotransmitters into synaptic vesicles to enhance quantal signals. Consistent with this, we observed an up-regulation of vGluT1 (by increased synthesis and decreased degradation) and vGluT2 (by increased synthesis) during TTX-induced up-scaling (*Figure 5*). During BIC-induced down-scaling, on the other hand, we found a decrease in the synthesis of vGluT1 (Figure 4), which could, in principle, lower vesicular neurotransmitter levels, leading to decreased mEPSC amplitudes. Besides the glutamate transporters, many other presynaptic proteins were regulated upon BIC or TTX treatment, suggesting that additional presynaptic mechanisms contribute to homeostatic scaling. Overall, most synaptic proteins exhibited a decrease in synthesis during homeostatic down-scaling (*Figure 4*). Correspondingly, we found a significant over-representation of synaptic GO terms in the group of proteins that showed decreased synthesis or decreased synthesis together with increased degradation during homeostatic down-scaling (including synapse, excitatory synapse, glutamatergic synapse, postsynapse, presynapse, and synaptic vesicle; *Supplementary file 1*). This finding is consistent with the concept of synaptic down-scaling decreasing neuronal network activity. Only a few synaptic proteins were up-regulated during BIC-induced down-scaling. The strongest up-regulation was observed for Bdnf and Pcdh8, which exhibited increased synthesis as well as decreased degradation. Bdnf is released in a calcium- and activity-dependent manner, and was one of the first proteins implicated in homeostatic scaling (*Rutherford et al., 1998*). *Rutherford et al. (1998)* showed that Bdnf is required for and negatively regulates homeostatic up-scaling. Pcdh8 (also named Arcadlin) is an activity-induced protocadherin that binds to N-cadherin to promote its endocytosis and thereby weaken synaptic connections (*Arikkath and Reichardt, 2008*; *Yasuda et al., 2007*). Our data suggest that the expression of Pcdh8 might bidirectionally regulate the strength of synaptic connections during homeostatic down-scaling (through increased synthesis and decreased degradation of Pcdh8) and homeostatic up-scaling (through decreased synthesis of Pcdh8).

During TTX-induced up-scaling, many synaptic proteins were up-regulated by increased synthesis and/or decreased degradation. For instance, increased protein synthesis was observed for GluA1, but not for GluA2 and GluA3. GluA2-lacking receptors have a higher $Ca^{2+}$ permeability, channel conductance and open probability than GluA2-containing receptors (*Burnashev et al., 1992*; *Oh and Derkach, 2005*; *Swanson et al., 1997*), and an increased contribution of GluA2-lacking AMPARs has been observed upon synaptic strengthening induced by long-term activity blockade by

TTX and APV (*Sutton et al., 2006*; *Thiagarajan et al., 2005*). Nitric oxide synthase 1 (Nos1) — a protein that is required for long-term potentiation (*Haley et al., 1992*; *Hardingham et al., 2013*; *Schuman and Madison, 1991*) — is another interesting candidate that is up-regulated by increased synthesis as well as by decreased degradation during TTX-induced up-scaling. Hence, our data suggest that Nos1 and NO signaling is also regulated by homeostatic up-scaling and might be a link between the presynaptic and postsynaptic mechanisms underlying homeostatic scaling. Among the proteins that were down-regulated by decreased synthesis, we found an over-representation of (among others) proteins associated with GABA-ergic synapses, which makes sense for a system that is in the process of up-scaling. Together, these changes accomplish a proteomic remodeling at the synapses that might strengthen the synaptic connections.

Temporal analysis revealed that many proteins showed transient changes in synthesis and/or degradation during homeostatic scaling (*Figures 7,8*). Proteins that were only regulated at the first time point or that showed decreasing strength of regulation (e.g. Bdnf during down-scaling; *Figure 8B*) might play a role in the initiation of homeostatic scaling, whereas proteins that were permanently regulated (such as PSD-95 during down-scaling; *Figure 8B*) might be involved in maintenance of the scaling.

In addition to the protein candidates highlighted above, our comprehensive dataset contains hundreds of proteins that showed changes in synthesis and/or degradation during homeostatic up- or down-scaling. The use of the 'semi-heavy' internal standard enabled us to correct for the experimental and technical variation introduced during sample preparation and LC-MS acquisition. Therefore, we obtained a high-precision dataset that allowed us to quantify even small changes in protein synthesis and degradation with statistical significance; small changes are often missed with label-free or label-free-like (separate evaluation of isotopic channels without internal normalization) experimental designs. We believe that small changes in protein synthesis and/or degradation (detected in the whole-cell lysate) might be biologically relevant as they could have a strong local impact either in different neuron types or at specific sub-cellular locations. In support of our data, many of these regulated proteins were previously implicated in synaptic plasticity. In addition, we discovered regulation of proteins that were not yet implicated in homeostatic scaling or synaptic plasticity, which are promising candidates for future studies.

## Materials and methods

### Preparation and maintenance of primary cultured hippocampal cells

Dissociated hippocampal neurons were prepared and maintained as previously described (*Aakalu et al., 2001*). Briefly, hippocampi from postnatal day one rat pups (strain Sprague-Dawley, RRID:RGD_734476) were dissected and dissociated by papain and plated onto poly-D-lysine-coated Petri dishes (MatTek, Ashland, MA). Cultured cells were maintained in Neurobasal-A medium (Invitrogen, Carlsbad, CA) supplemented with B-27 (Invitrogen) and Glutamax (Invitrogen) at 37°C.

### Dynamic SILAC experiment

After 18–19 days in vitro (DIV), the culture medium was exchanged with a medium that was depleted of arginine and lysine (customized; Invitrogen) and was supplemented with 'heavy' isotopically labeled arginine (R10; Thermo, Waltham, MA) and lysine (K8; Thermo), resulting in a final percentage of 80% heavy arginine/lysine and 20% remaining light arginine/lysine. Visual inspection of the cells in pilot experiments revealed improved cell viability when a thin layer of initial medium remained on the cells during the medium change compared to a complete medium exchange. Together with the medium change, 20 μM BIC, 1 μM TTX or no drug (Ctrl) was added. The cells were harvested after 1, 3 or 7 days of the treatment and heavy SILAC pulse. Prior to harvest, all cells were visually inspected under the microscope. Neurons that showed fragmented dendrites or dishes with low cell density were excluded. The cells were washed with ice-cold DPBS (Invitrogen) supplemented with protease inhibitor (cOmplete EDTA-free, Roche, Basel, Switzerland), then scraped and pelleted by centrifugation. The cell pellets were lysed in lysis buffer (8 M urea, 200 mM Tris/HCl [pH 8.4], 4% CHAPS, 1 M NaCl, cOmplete EDTA-free protease inhibitor) using a pestle and sonication for 4 × 30 s at 4°C. The lysates were incubated with Benzonase (1 μL of a ≥ 250 units/mL stock solution; Sigma, St. Louis, MO) for 10 min and cleared by centrifugation for 5 min at 10,000 x g. Protein

concentration was determined via BCA assay (Thermo). Three independent biological replicates were performed from each different preparation.

## Preparation of semi-heavy neurons as internal standard

Primary hippocampal neurons were prepared and plated as described above. Six hours after plating, the medium was fully exchanged with a medium that was depleted of arginine and lysine (customized; Invitrogen) and supplemented with 'semi-heavy' isotopically labeled arginine (R6; Thermo) and lysine (K4; Thermo). After 21 DIV, the cells were harvested and lysed as described above. Semi-heavy lysates from different culture dishes were merged to create one master mix ('semi-heavy internal standard'; IS). Protein concentration was determined via BCA assay (Thermo).

## Sample preparation for MS analysis

Each 50 µg (protein amount) sample of the dynamic SILAC lysates was mixed with 25 µg of semi-heavy internal standard lysate ('triple SILAC' samples) and further processed for MS analysis. In addition, 25 µg samples of the dynamic SILAC lysates without internal standard ('double SILAC' samples) were also processed for MS analysis. Note, in one biological replicate, 'double SILAC' samples were only prepared for the untreated control samples (sample overview in *Supplementary file 3*). The proteins were digested according to the 'Filter-Aided Sample Preparation' (FASP) protocol as described by *Wiśniewski et al. (2009)*. After digestion, the 'double SILAC' samples were desalted using C18 StageTips (*Rappsilber et al., 2007*). The digested 'triple SILAC' samples were fractionated into four fractions using strong-cation-exchange (SCX) StageTips (*Rappsilber et al., 2007*). All fractions were desalted using C18 StageTips. Samples were dried by vacuum centrifugation and stored at −20˚C until LC-MS analysis.

## LC-MS/MS analysis

The dried peptide samples were reconstituted in 5% acetonitrile (ACN) with 0.1% formic acid (FA) and subsequently loaded using a nano-HPLC (Dionex U3000 RSLCnano) onto a PepMap100 loading column (C18, L = 20 mm, 3 µm particle size, Dionex) and washed with loading buffer (2% ACN, 0.05% trifluoroacetic acid (TFA) in water) for 6 min at a flow rate of 6 µL/min. Peptides were separated on a PepMap RSLC analytical column (C18, L = 50 cm, <2 µm particle size, Dionex) by a gradient of phase A (water with 5% v/v dimethylsulfoxide [DMSO] and 0.1% FA) and phase B (5% DMSO, 15% water and 80% ACN v/v/v). The gradient was ramped from 4% B to 48% B in 178 min at a flow rate of 300 nL/min. All solvents were LC-MS grade and purchased from Fluka. Eluting peptides were ionized online using a Nanospray Flex ion source (Thermo Scientific) and analyzed either in a QExactive Plus (Thermo Scientific) or in a Lumos Fusion (Thermo Scientific) mass spectrometer in data-dependent acquisition mode. The full parameter sets are listed in *Supplementary file 2*. All fractions of the 'triple SILAC' samples were measured in technical duplicates. For 'double SILAC' samples, one technical replicate was measured.

## Database searches

Raw data were analyzed with MaxQuant (version 1.6.2.3; RRID:SCR_014485;*Cox and Mann, 2008*; *Tyanova et al., 2016*) using customized Andromeda parameters (see *Supplementary file 3*). For all searches, spectra were matched to a *Rattus norvegicus* database downloaded from uniprot.org (reviewed and unreviewed; RRID:SCR_002380) and a contaminant and decoy database. Carbamidomethylation of cysteine residues was set as a fixed modification. Protein-N-terminal acetylation and methionine oxidation were set as variable modifications. A false discovery rate (FDR) of 1% was applied at the peptide-spectrum-match (PSM) and protein level. To assess the rate of re-incorporation of 'light' arginine and lysine into nascent proteins during the' heavy' pulse ('recycling rate'), the 'double SILAC' samples were analyzed with multiplicity set to one, and heavy arginine (R10) and heavy lysine (K8) were set as additional variable modifications. For precise quantification of nascent and pre-existing proteins, the 'triple SILAC' samples were searched with a multiplicity of 3 (light [K0, R0], semi-heavy [K4, R6], and heavy [K8, R10]). All MaxQuant results were filtered to remove contaminants and decoys. If not stated otherwise, only unique peptides were included in downstream analysis. All proteomics data associated with this manuscript have been uploaded to the PRIDE repository (RRID:SCR_003411) (*Vizcaíno et al., 2013*) with accession number PXD016004.

## Bioinformatic processing and data analysis

### Re-incorporation of light arginine and lysine into nascent proteins

Peptides containing two arginine and/or lysine residues (due to a missed tryptic cleavage site) were used for the analysis. The ratio of the following combinations was calculated for each sample (except $t_0$ samples) based on the number of detections: 'light-heavy' and 'heavy-heavy'. The probabilities of incorporation of light or heavy arginine or lysine into nascent proteins were calculated as described by *Dörrbaum et al. (2018)*. For each biological replicate, the average heavy incorporation probability was calculated and used as a 'recycling correction factor' for the 'triple SILAC' data. These recycling correction factors were used to calculate the fraction of pre-existing peptides from the fraction of 'light' peptides and the fraction of newly synthesized peptides from the fraction of heavy peptides.

### Protein half-life calculation

Protein half-lives were determined as described by *Dörrbaum et al. (2018)* with minor modifications. Using the peptide results of the 'triple SILAC' data, the fractions of the remaining light peptides (%L) were calculated for each measurement and each peptide based on the H/L ratios. For $t_0$ samples, %L was set to 1 if the peptide was only detected in its 'light' form. The fraction of light peptide was converted into the fraction of pre-existing peptide (%old) using the above-described correction factors, which correct for the incorporation of light Arg/Lys into newly synthesized proteins. Resulting negative values were excluded from further analysis. Peptides were subsequently filtered within biological replicates. Only unique peptides that were quantified at all four time points ($t_0$, 1d, 3d, 7d) and with a mean %old >0.9 at $t_0$ were considered for further analysis. In a few cases, two protein groups were merged (in order to rescue peptides) and only peptides unique for the merged group were further considered. For downstream analysis, only protein groups in which at least one peptide fulfilled the above criteria in each biological replicate were used. Peptide outliers as described by *Dörrbaum et al. (2018)* were removed. For each protein and each condition (BIC, TTX, Ctrl), the filtered peptide data (%old over time) were natural log transformed and fitted by a linear function. Protein half-lives ($t_{1/2}$) were calculated using the rate constant (negative value of the slope of the fit).

### Quantification of nascent and pre-existing proteins over the entire time course

Nascent proteins were quantified using the H/M ratios, and pre-existing proteins were quantified using the L/M ratios computed by MaxQuant. MaxQuant peptide results from the 'triple SILAC' data were filtered for decoys and contaminants. For each protein in each sample, the sample-to-IS ratio ([H+L]/M) was calculated. To correct for slight inaccuracies during IS spike-in, all samples were scaled to the same median sample-to-IS ratio. The 'recycling correction factors' (described above) were used to correct for re-incorporation of light Arg/Lys into nascent proteins during the heavy pulse, and to convert the scaled H/M signals into nascent protein signals and the L/M signals into pre-existing protein signals. Resulting negative values were excluded from further analysis. For the quantitative comparison between different conditions (treatments vs untreated control), only unique peptides that were quantified in both the treatment and the corresponding control sample (treatment–control peptide pairs) were used. Only proteins with at least one treatment–control peptide pair per time point in all biological replicates were used for downstream analysis. For statistical analysis of changes in protein synthesis or degradation, a linear mixed-effect model was applied (*Bates et al., 2014*; *West and Galecki, 2011*). Treatment was defined as the fixed effect of interest, time point was set as an additional fixed effect, and biological replicate as well as peptide identity nested into biological replicate were set as random effects. Benjamini-Hochberg correction was applied to correct for multiple comparisons.

### Quantification of nascent and pre-existing proteins for individual time points

Nascent and pre-existing proteins were quantified as described above with minor adaptions. Only proteins with at least one treatment–control peptide pair at the time point of interest in all biological

replicates were used for downstream analysis. In the linear mixed-effect model, treatment was defined as the fixed effect of interest, and biological replicate as well as peptide identity nested into biological replicate were set as random effects. Benjamini-Hochberg correction was applied to correct for multiple comparisons.

## Quantification of relative protein abundance over the entire time course

Relative protein abundance was quantified using the sum of H/M and L/M ratios computed by Max-Quant. For each peptide in each sample, the sample-to-IS ratio ([H+L]/M) was calculated. To correct for slight inaccuracies during IS spike in, all samples were scaled to the same median sample-to-IS ratio. For the quantitative comparison between different conditions (treatments vs untreated control), only unique peptides that were quantified in both the treatment and the corresponding control sample (treatment–control peptide pairs) were used. Only proteins with at least one treatment–control peptide pair per time point in all biological replicates were used for downstream analysis. For statistical analysis of changes in relative protein abundance, a linear mixed-effect model was applied. Treatment was defined as the fixed effect of interest, time point was set as additional fixed effect, and biological replicate and peptide identity nested into biological replicate were set as random effects. Benjamini-Hochberg correction was applied to correct for multiple comparisons.

## GO over-representation analysis

GO over-representation analysis was performed using the Gene List Analysis tool of the Panther Classification System (RRID:SCR_015893) (*Mi et al., 2019*). GO term over-representation was tested for all significantly regulated proteins by Fisher's exact test. All quantified proteins were used as reference data set. Significantly over-represented GO terms ($p<0.05$; FDR corrected) are shown in **Supplementary file 1**.

## Synaptic proteins

Synaptic proteins shown in **Figures 4,5** were selected according to UniProt subcellular location annotation and GO cellular component annotation.

## Clustering of temporal profiles of pre-existing and nascent proteins

Regulation of pre-existing and nascent proteins during BIC or TTX treatment were quantified at individual time points (1, 3 and 7 days) as described above. Fold changes were calculated for proteins that were significantly regulated (1% FDR) at individual time points. Fold changes were set to zero if the protein was not significantly regulated at the respective time point. Proteins were then clustered usingtheir temporal profiles of pre-existing and newly synthesized proteins (fold changes over time) by kmeans clustering (**Figure 8—figure supplement 1**; **Figure 8—figure supplement 2**). Clusters that shared similar profiles were manually assigned to upper-level groups.

## Enrichment analysis of uORF-containing transcripts

To identify transcripts containing uORFs, only genes with annotated 5'UTRs were considered. A string-match algorithm was used to identify sequences within annotated 5'UTRs that are flanked by a canonical in-frame start and stop codon. Only sequences with a minimum length of three codons were considered as uORFs. A protein group was classified as 'uORF-containing' if at least one protein was encoded by a uORF-containing gene.

## **Electrophysiological recordings**

Whole-cell recordings were performed in hippocampal neurons (prepared as described above) held at $-70$ mV in voltage clamp mode. To induce homeostatic scaling, the neurons were treated with BIC (20 µM) or TTX (1 µM) for 7 days starting on DIV 11–13. Miniature excitatory postsynaptic currents (mEPSCs) were recorded for at least 10 min in extracellular solution containing 140 mM NaCl, 3 mM KCl, 10 mM HEPES, 2 mM CaCl$_2$, 1 mM MgSO$_4$, 15 mM glucose and 1 µM TTX (pH 7.4). For recordings of BIC-treated cells, also 20 µM BIC was added. Recording pipettes (resistances 6–10 MΩ) contained 120 mM potassium gluconate, 20 mM KCl, 10 mM HEPES, 2 mM MgCl$_2$, 0.1 mM EGTA, 2 mM Na$_2$-ATP, and 0.4 mM Na$_2$-GTP (300 mOsm/L [pH 7.2]). Data were analyzed offline

in Clampfit. mEPSC events were screened with an amplitude threshold of >4 pA and an exponential decay. Statistics were conducted using unpaired t-tests.

## Quantification of nascent proteins by AHA pulse labeling and western blot analysis

Primary hippocampal cultures were prepared as described above. After 18–19 DIV, the cells were incubated with 20 µM BIC, 1 µM TTX or no drug (Ctrl) for 24 hr. During the last 2 hr of the treatment, the cells were incubated in methionine-free Neurobasal-A (customized; Invitrogen) supplemented with 4 mM azidohomoalanine (AHA; in-house synthesized). In methionine control samples, the medium was supplemented with 4 mM methionine (Sigma) instead of AHA. After metabolic labeling, the cells were washed with ice-cold PBS (Invitrogen) and scraped in PBS supplemented with 0.5% SDS, 0.5% Triton-X100 and protease inhibitor cocktail (Roche). The lysates were sonicated for $4 \times 30$ s at 4°C, heated to 95°C for 10 min, and cleared by centrifugation at 10,000 x g for 5 min. The lysates were diluted with PBS to decrease the SDS concentration below 0.1%. Protein concentration was determined by BCA assay (Thermo Scientific) according to the manufacturer's instructions. Equal protein amounts were used for click reaction. The lysates were incubated with triazole ligand tris([1-benzyl-1H-1,2,3-triazol-4-yl]methyl)amine (TBTA; 300 µM), biotin alkyne tag (50 µM) and Cu(I)Br (80 µg/mL) overnight at 4°C in the dark. Biotinylated proteins were then separated by electrophoresis and immunoblotted with anti-biotin antibodies (1:1000, Cell Signaling Technology, Ref: 5597). Three independent biological replicates from different preparations were performed.

## Western blot analysis

Primary hippocampal cultures were prepared as described above. After 18–19 DIV, the cells were incubated with 20 µM BIC, 1 µM TTX or no drug (Ctrl) for 1, 3 or 7 days as indicated, subsequently washed with ice-cold PBS (Invitrogen) and scraped in PBS supplemented with 0.5% SDS, 0.5% Triton-X100, protease inhibitor cocktail (Roche), Halt phosphatase inhibitor (Thermo Scientific) and benzonase (2 µL/mL, ≥250 units /µL, Sigma). The lysates were cleared by centrifugation at 17,000 x g for 15 min. Protein concentration was determined by BCA assay (Thermo Scientific). Equal protein amounts were used for western blot analysis. Proteins were separated by electrophoresis and immunoblotted with antibodies against Bdnf (1:400, Santa Cruz Biotechnology, Ref: sc-546), Homer1 (1:1000, Synaptic Systems, Ref: 160003), GluA1 (1:1000, Synaptic Systems, Ref: 182003), Synaptopodin (1:1000, Synaptic Systems, Ref:163002), Camk1g (1:2000, Abcam, Ref: ab227209), β-actin (1:3000, Sigma, Ref: A5316), eIF2α (1:5000, Cell Signaling Technology, Ref: L57A5) and p-eIF2α (1:2500, Invitrogen Ref: 44728G). For western blots against eIF2α and p-eIF2α, phosphatase inhibitor was added to all solutions. Three biological replicates were performed.

## Proteasome activity assay

Primary hippocampal cultures were prepared as described above. After 18–19 DIV, the cells were incubated with 20 µM BIC, 1 µM TTX or no drug (Ctrl) for 7 days. Subsequently, cells were washed with ice-cold PBS (Invitrogen) and scraped in 50 mM HEPES (pH 7.4) supplemented with 50 mM NaCl, 5 mM EDTA (Sigma), 10 µM leupeptin (Sigma), 1 µg/mL pepstatin-A (Sigma), and 1 mM phenylmethylsulfonyl fluoride (Sigma). Lysates were prepared by freeze-thaw cycles and Dounce homogenization, and cleared by centrifugation at 3000 x g for 10 min at 4°C. Protein concentration was determined by BCA assay (Thermo Scientific) and equal protein amounts were used for the proteasome activity assay. Proteasome chymotrypsin-like activity was determined with the synthetic fluorogenic peptide N-Suc-LLVY-MCA (Sigma) in a fluorescence microplate reader (excitation, 400 nm; emission, 505 nm). Fluorescence was measured every 2 min for 4 hr and data were evaluated in the linear range. Three biological replicates were performed.

## Acknowledgements

We thank I Bartnik, N Fuerst, A Staab and C Thum for the preparation of primary cell cultures and F Rupprecht for MS maintenance and assistance with data acquisition. Analysis of uORF-containing genes was performed by A Biever, C Glock and G Tushev. EMS is funded by the Max Planck Society and DFG CRC 1080: Molecular and Cellular Mechanisms of Neural Homeostasis and DFG CRC 902: Molecular Principles of RNA-based Regulation. This project has received funding from the European

Research Council (ERC) under the European Union's Horizon 2020 research and innovation programme (grant agreement No 743216).

# Additional information

## Funding

| Funder | Grant reference number | Author |
|---|---|---|
| Max Planck Society | | Aline Ricarda Dörrbaum<br>Beatriz Alvarez-Castelao<br>Belquis Nassim-Assir<br>Julian D Langer<br>Erin M Schuman |
| Horizon 2020 Framework Programme | 743216 | Aline Ricarda Dörrbaum<br>Beatriz Alvarez-Castelao<br>Belquis Nassim-Assir<br>Julian D Langer<br>Erin M Schuman |
| DFG | CRC 1080 | Erin M Schuman |
| DFG | CRC 902 | Erin M Schuman |

The funders had no role in study design, data collection and interpretation, or the decision to submit the work for publication.

## Author contributions

Aline Ricarda Dörrbaum, Conceptualization, Data curation, Formal analysis, Validation, Investigation, Visualization, Methodology, Project administration, Writing—original draft, Writing—review and editing; Beatriz Alvarez-Castelao, Conceptualization, Formal analysis, Investigation; Belquis Nassim-Assir, Formal analysis, Investigation; Julian D Langer, Conceptualization, Formal analysis, Supervision; Erin M Schuman, Conceptualization, Supervision, Funding acquisition, Project administration, Writing—review and editing

## Author ORCIDs

Aline Ricarda Dörrbaum https://orcid.org/0000-0002-1178-7078

Julian D Langer https://orcid.org/0000-0002-5190-577X

Erin M Schuman https://orcid.org/0000-0002-7053-1005

## Ethics

Animal experimentation: All hippocampal neurons were derived from P0 (postnatal day 0) CD Crl: CD (Sprague Dawley) rat pups (both male and female, RRID: RGD_734476). Pregnant females from timed matings were delivered from Charles River Laboratories as SPF (Specific-Pathogen Free) animals and housed in the institute's animal facility for one week at a 12/12 hour light dark cycle with food and water ad libitum until the litter was born. Pups were sacrificed by decapitation with sharp scissors. The housing and sacrificing procedures involving animal treatment and care were conducted in conformity with the institutional guidelines that are in compliance with national and international laws and policies (DIRECTIVE 2010/63/EU; German animal welfare law; FELASA guidelines). The animals were euthanized according to annex 2 of § 2 Abs. 2 Tierschutz-Versuchstier-Verordnung. Animal numbers were reported to the local authority (Regierungspräsidium Darmstadt).

## Decision letter and Author response

Decision letter https://doi.org/10.7554/eLife.52939.sa1

Author response https://doi.org/10.7554/eLife.52939.sa2

## Additional files

### Supplementary files

- Supplementary file 1. GO term over-representation of proteins with distinct regulation.
- Supplementary file 2. MS settings.
- Supplementary file 3. Sample overview and MaxQuant search parameters.
- Transparent reporting form

### Data availability

All proteomics data associated with this manuscript have been uploaded to the PRIDE repository (Vizcaino et al., 2013) with accession number PXD016004.

The following dataset was generated:

| Author(s) | Year | Dataset title | Dataset URL | Database and Identifier |
|---|---|---|---|---|
| Langer J | 2019 | Neuronal proteome dynamics during homeostatic scaling. | https://www.ebi.ac.uk/pride/archive/projects/PXD016004 | PRIDE , PXD016004 |

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
