## [Decision Letter]

Thank you for submitting your article "Proteome dynamics during homeostatic scaling in cultured neurons" for consideration by *eLife*. Your article has been reviewed by two peer reviewers, and the evaluation has been overseen by Catherine Dulac as the Senior Editor. The reviewers have opted to remain anonymous.

The reviewers have discussed the reviews with one another and the Reviewing Editor has drafted this decision to help you prepare a revised submission.

In this manuscript, the authors have quantified changes in protein synthesis and degradation as well as changes in protein turnover and abundance in hippocampal neuronal cultures during homeostatic scaling induced by neuronal activity. The authors demonstrate that a large percentage of the neuronal proteome changes during to synthesis and/or degradation during both homeostatic up- and down-scaling. Especially interesting, and perhaps surprising, is that more than half of the quantified synaptic proteins were regulated. These included both pre- and post-synaptic proteins that have diverse functions.

However, the evaluation indicated several reservations. There are deficits raised in the experiments and the description of the results that warrant improvements. The first is the need for additional validation of the mass spectrometry results. A second problem is the need to emphasize the main findings in a clearer fashion, which will require a reorganization of the paper. Both reviewers felt these concerns could be addressed in a two month period. Their detailed comments are listed below.

Essential revisions:

1) The major concern is that the authors are assuming that endogenous proteins are all light labeled and that all newly synthesized proteins are heavy labeled. The potential problem is that not every protein will have abundant arginine/lysine amino acids and that no label is 100% efficient. How do the authors know that they have 100% efficiency in labeling? The authors could address this issue in either of two ways. One would be to double label with SILAC and AHA (i.e., QuaNCAT). The other way would be to medium label, followed by a heavy label for the 24 hours, 3 days, 7 days, etc. In either case, there would be a coincidence detection and the authors could make a more direct comparison between two groups of labeled proteins.

2) The experiments are generally well performed however, the conclusions are rather complicated and in some cases vague – often leaving the reader wondering what can be learned from all these measurements. The manuscript contains a very large number of figures (nine) and the flow of narrative is rather winding and the experiments do not flow well together. Overall, this reviewer is very fond of these type of projects and many of the compelling previous studies from the Schuman lab. However, this manuscript reads as it was pieced together from extra datasets that were left out of previous publications. Many modern and powerful strategies have been used but very tough to interpret most of the results, save for a few controls (such as BDNF). The take away message it just too soft.

---

## [Author Response]

Essential revisions:1) The major concern is that the authors are assuming that endogenous proteins are all light labeled and that all newly synthesized proteins are heavy labeled. The potential problem is that not every protein will have abundant arginine/lysine amino acids and that no label is 100% efficient. How do the authors know that they have 100% efficiency in labeling?

For our analysis, we only use tryptic peptides (with up to 2 missed cleavages) that contain at least one arginine or lysine. More than 99% of the proteins contained in the UniProt database for *Rattus norvegicus* contain at least one arginine or lysine, making almost the entire proteome accessible for this approach. Based on the measured peptide precursor masses, we can unambiguously distinguish heavy from light peptide species. This is a well-established approach in the proteomics community. All pre-existing proteins (proteins that were produced before the heavy SILAC labeling) were ‘light labeled’, as the cells were only exposed to natural light, but not heavy amino acids before. Natural (light) peptides also contain natural heavy isotopes, however, due to the low abundance of natural heavy isotopes, the interference with heavy or semi-heavy labeled peptides (resulting from SILAC labeling) is negligible (see isotopic patterns in Figure 1—figure supplement 1). Upon the change to the heavy SILAC medium, mainly heavy arginine and lysine were incorporated into nascent proteins. However, the medium contained approximately 20% of remaining light arginine/lysine and additional light arginine/lysine might have become available by degradation of pre-existing proteins. To correct for the potential incorporation of light arginine/lysine into nascent proteins during the heavy SILAC pulse, we developed a correction factor. Based on peptides that contained two arginine/lysine residues (due to one missed tryptic cleavage site), we calculated the probability of the incorporation of light or heavy arginine/lysine, respectively. The procedure is described in more detail in the Materials and methods section.

The authors could address this issue in either of two ways. One would be to double label with SILAC and AHA (i.e., QuaNCAT).

The double labeling with AHA and SILAC (‘QuaNCAT’) is a powerful approach to quantify nascent proteins after short labeling times, when the SILAC label is not quantifiable without prior enrichment. However, in our experiments, newly synthesized proteins are labeled for longer durations of 24 hours, 3 days and 7 days. We and others previously showed that 1 day SILAC labeling in primary neuronal cultures is sufficient for accurate quantification of heavy and light peptide counterparts (Cohen et al., 2013; Mathieson et al., 2018; Dörrbaum et al., 2018).

The other way would be to medium label, followed by a heavy label for the 24 hours, 3 days, 7 days, etc. In either case, there would be a coincidence detection and the authors could make a more direct comparison between two groups of labeled proteins.

We don’t understand the advantage of a ‘semi-heavy’ (‘medium’) pulse prior to the ‘heavy’ pulse. We hope that the above explanation about how all proteins are light-labeled from the outset, might clarify things for the reviewer.

2) The experiments are generally well performed however, the conclusions are rather complicated and in some cases vague – often leaving the reader wondering what can be learned from all these measurements. The manuscript contains a very large number of figures (nine) and the flow of narrative is rather winding and the experiments do not flow well together. Overall, this reviewer is very fond these type of projects and many of the compelling previous studies from the Schuman lab. However, this manuscript reads as it was pieced together from extra datasets that were left out of previous publications. Many modern and powerful strategies have been used but very tough to interpret most of the results, save for a few controls (such as BDNF). The take away message it just too soft.

We are surprised that the reviewer finds this manuscript reads as if it was pieced together from left-over data sets. This is absolutely not the case. The aim of the project was singular: to quantify the proteome dynamics that underlie homeostatic scaling in neurons. To this end, we used dynamic SILAC labeling together with mass spectrometry to quantify the changes in protein synthesis and degradation as well as resulting changes in protein turnover or protein abundance for several thousand neuronal proteins. The SILAC-MS data, which is the first dataset that comprehensively describes the changes in protein synthesis, degradation, turnover and abundance during homeostatic scaling, is the core of the manuscript. All additional experiments were performed to validate, support or explain the SILAC-MS results, as outlined below:

– Electrophysiological recordings were performed to confirm that BIC and TTX treatments induced homeostatic down- or up-scaling, respectively, over the entire duration of the experiment.

– AHA labeling and western blot analysis was performed to validate the finding that a large fraction of proteins exhibited decreased synthesis rates during homeostatic scaling.

– Following up on the observation that protein synthesis was reduced during homeostatic scaling, western blot analysis was performed to demonstrate that eIF2α was phosphorylated during homeostatic scaling. Phosphorylation of eIF2α inhibits protein synthesis and is hence one underlying mechanism that may lead to the decrease in protein synthesis observed in the SILAC-MS data.

– Based on the SILAC-MS data we also found that a large proportion of proteins exhibited reduced degradation rates during homeostatic scaling. Following up on this observation, we analyzed ongoing proteasomal degradation (proteasome activity assay) and autophagy (based on the SILAC-MS data) to find that neither proteasome activity nor ongoing autophagy was impaired during homeostatic scaling.

– In addition, we now included western blot analyses to verify our SILAC-MS results for some candidate proteins.

We nevertheless take the reviewer’s comment to indicate that the original version of the paper was very dense- we agree. To improve the flow of the manuscript, we condensed and restructured the Results and Discussion sections to make the take-home messages more clear. We also wish to emphasize, though, that we chose *eLife* as the preferred journal for publication because it allows for the publication and exploration of complete datasets.